# EOP: Unlocking Superior Problem Solving in Small LLMs

## Abstract

Small language models, referred to as LLMs with fewer than 10 billion parameters in this work, face critical challenges in problem-solving tasks, often achieving less than 10% accuracy, highlighting the urgent need for effective solutions. While much of the existing research has focused on enhancing the performance of larger models like GPT, an important question remains: Can techniques developed for large models be adapted effectively for smaller ones? Moreover, is it possible to improve these smaller models to the point where they rival, or even outperform, larger models such as GPT-4 in problem-solving tasks?

In this paper, we introduce Evaluation-Oriented Problem-Solving (EOP), a novel framework aimed at enhancing the problem-solving capabilities of small LLMs. Our approach significantly boosts the performance of these models, achieving a 2% higher accuracy on Python Puzzles compared to standard GPT-4 and a 27% improvement over state-of-the-art prompting methods using GPT-4 in the Game of 24. Beyond these results, EOP also demonstrates notable accuracy improvements on other tasks. These findings suggest that, with the appropriate strategies, small LLMs can achieve substantial performance gains in problem-solving, challenging the prevailing notion that scaling model size is the primary path to improvement.

## 1 Introduction

Large Language Models (LLMs) have advanced rapidly, demonstrating strong performance across various intelligent tasks such as commonsense reasoning (Hendrycks et al., 2020; Wang, 2018), question-answering (Joshi et al., 2017), scientific knowledge acquisition (Clark et al., 2018; Cobbe et al., 2021), and programming (Chen et al., 2021). The open-source community has also contributed to this growth, offering models of diverse sizes and capabilities. Small LLMs, typically with fewer than 10 billion parameters, are gaining traction due to their advantages in cost and energy efficiency, lower hardware requirements for self-deployment, and reduced domain adaptation costs, all while maintaining relatively high performance on various benchmarks such as reading comprehension and common-sense understanding (Bansal et al., 2024; Javaheripi et al., 2023; Dubey et al., 2024).

Despite their strengths in natural language understanding, LLMs face significant challenges in problem-solving (Wei et al., 2022; Yao et al., 2024; Besta et al., 2024; Dubey et al., 2024). Even advanced models like GPT-4 (OpenAI, 2024a) struggle, consistently scoring poorly on various benchmarks (OpenAI, 2024b), with smaller models performing even worse due to their limited parameters. Experiments on three problem-solving tasks revealed this gap: GPT-4 scored just 3.0% on *Game of 24* (Yao et al., 2024) and 31.1% on *Python Puzzles* (Schuster et al., 2021), while smaller models fared even worse, as shown in Figure 1.

Various approaches, such as Chain-of-Thought (CoT) (Wei et al., 2022), Tree-of-Thought (ToT) (Yao et al., 2024), Graph-of-Thought (GoT) (Besta et al., 2024), and Meta-prompting (Suzgun & Kalai, 2024), have been proposed to enhance the performance of GPT models. These prompting techniques significantly elevate the reasoning capabilities of GPT-4. In particular, Meta-prompting proves to be the most effective method, leading to substantial improvements in accuracy when applied to GPT-4, as demonstrated in Figure 1.

Given these contexts, we explored three critical questions to better understand the capabilities of small LLMs in reasoning tasks:

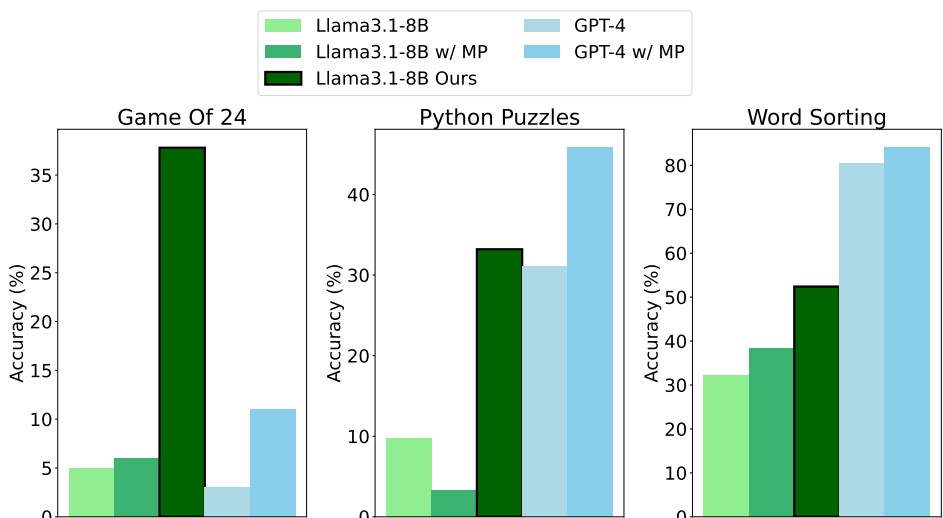

Figure 1: We compare the problem-solving accuracy across three tasks: **Game Of 24**, **Python Puzzles**, and **Word Sorting**, each with different *evaluation difficulty*. Python Puzzles uses an oracle Python interpreter to verify the answer, Game Of 24 is easy (evaluating the expression is equal to 24), and Word Sorting is the hardest (evaluating the list of words is in the correct order). Key observations: 1) GPT-4 (OpenAI, 2024a) performs poorly on some tasks, with Llama3.1-8B scoring even lower. 2) Meta-Prompting (MP) (Suzgun & Kalai, 2024) boosts GPT-4's performance but is less effective for small LLMs. 3) Our method enables small LLMs to outperform GPT-4 with MP on **Game Of 24** and surpass GPT-4 in **Python Puzzles**, demonstrating improvements across various tasks.

**Q1: Are current methods designed for large models applicable to small LLMs?** Although advanced prompting techniques, such as meta-prompting, effectively enhance reasoning in models like GPT-4, our experiments revealed no significant improvement in small LLMs. As shown in Figure 1, methods that succeed with larger models do not appear to be well-suited for their smaller counterparts. The reasons for this discrepancy remain unclear.

**Q2: Are small LLMs applicable in problem-solving tasks?** Despite their limitations, small LLMs are not entirely ineffective. When using with an oracle evaluator to cherry-pick the correct answer from multiple attempts, they showed improved accuracy, as presented in Figure 2. This indicates that small models can benefit from repeated trials, though they still struggle to identify correct solutions on their own.

**Q3: How can small LLMs be made effective in problem-solving tasks?** This question forms the core motivation of our work. We propose that the key to unlocking the problem-solving potential of small LLMs lies in developing a tailored approach that compensates for their weaknesses through evaluation-focused prompting techniques. By leveraging multiple trials and structured evaluations, small models can perform significantly better without requiring extensive fine-tuning. To develop a general framework that is effective across different tasks (i.e. *task-agnostic*), we chose three problem sets with ***varying levels of evaluation difficulty***: **Python Puzzles** (Schuster et al., 2021) requires solving Python problems with code comprehension. We have an oracle evaluator (a Python interpreter), making them the *simplest* to evaluate. **Game of 24** (Yao et al., 2024) requires using 4 numbers and 3 operators (each from $+ - \times \div$) to get an arithmetic result of 24. It is slightly more challenging but still *straightforward*, as it only requires checking if an expression equals 24. **Word Sorting** (Srivastava et al., 2023) requires output an alphabetically sorted list from a list of words. It poses a great challenge on evaluation, as evaluating the correctness involves checking whether each pair of words in a list is sorted character by character, which is considerably *more complex*.

In this paper, we make four key contributions:

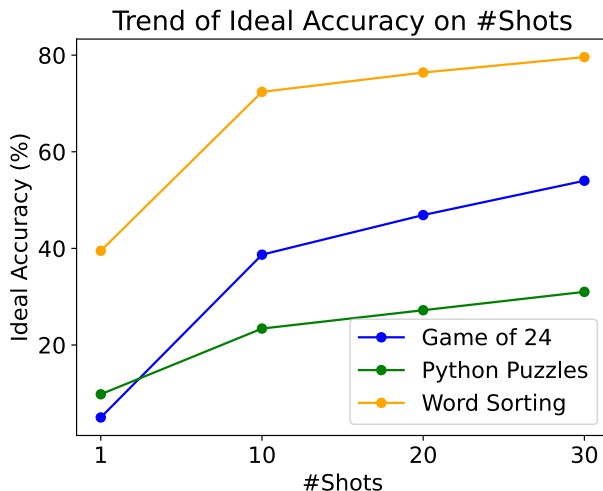

Figure 2: The *ideal* accuracy of the small LLM consistently improves as the number of attempts increases. *ideal* accuracy means utilizing an task-specific oracle evaluator to cherry-pick the correct answer from multiple trials. It demonstrates the potential of small LLMs: With a reliable evaluator, small models can achieve superior accuracy.

- **Understanding failure modes**: We identify why advanced prompting methods, such as meta-prompting, fail to improve the reasoning abilities of small LLMs. Specifically, small models struggle to process complex instructions and step-wise refining.

- **Emphasizing the role of evaluators**: We demonstrate the importance of robust evaluators in reasoning tasks. Our findings show that an oracle evaluator, when used with small LLMs, can dramatically improve their accuracy by enabling multiple attempts. This drives us to build a reliable evaluation scheme that can close the gap to the oracle evaluator.

- **Introducing a novel evaluation-based prompting method**: We propose a new, evaluation-focused prompting technique specifically designed for small LLMs. This method enhances reasoning performance, achieving a 2% higher accuracy on Python Puzzles compared to standard GPT-4 and a 27% improvement over state-of-the-art prompting methods using GPT-4 in the Game of 24, even without access to an oracle evaluator.

- **Addressing varying levels of evaluator expertise**: We explore how different reasoning tasks may require evaluators with varying levels of expertise. Our method adapts to these different cases, ensuring that small models can handle a wide range of tasks with the appropriate level of evaluation support.

## 2 RELATED WORK

Recent advancements in prompting methods have significantly enhanced the reasoning capabilities of LLMs. One of the earliest approaches, Chain-of-Thought (CoT) (Wei et al., 2022), guides LLMs by providing few-shot examples of reasoning steps or by appending instructions like "Let's think step by step" to encourage intermediate thinking. This method improves the likelihood of generating correct answers through guided reasoning. Building upon CoT, CoT with Self-Consistency (Wang et al., 2022) further refines this process by performing multiple reasoning attempts and selecting the final answer through result aggregation, which improves accuracy. Based on this method, more advanced techniques are developed:

**Tailored and Structured Thoughts:** Methods like Tree-of-Thoughts (ToT) (Yao et al., 2024), break down problems into subproblems, structuring reasoning as a tree. Graph-of-Thoughts (GoT) (Besta et al., 2024) extends this concept by transforming the tree into a graph, enhancing performance on more complex tasks like sorting. However, both methods require manual task-specific constructions.

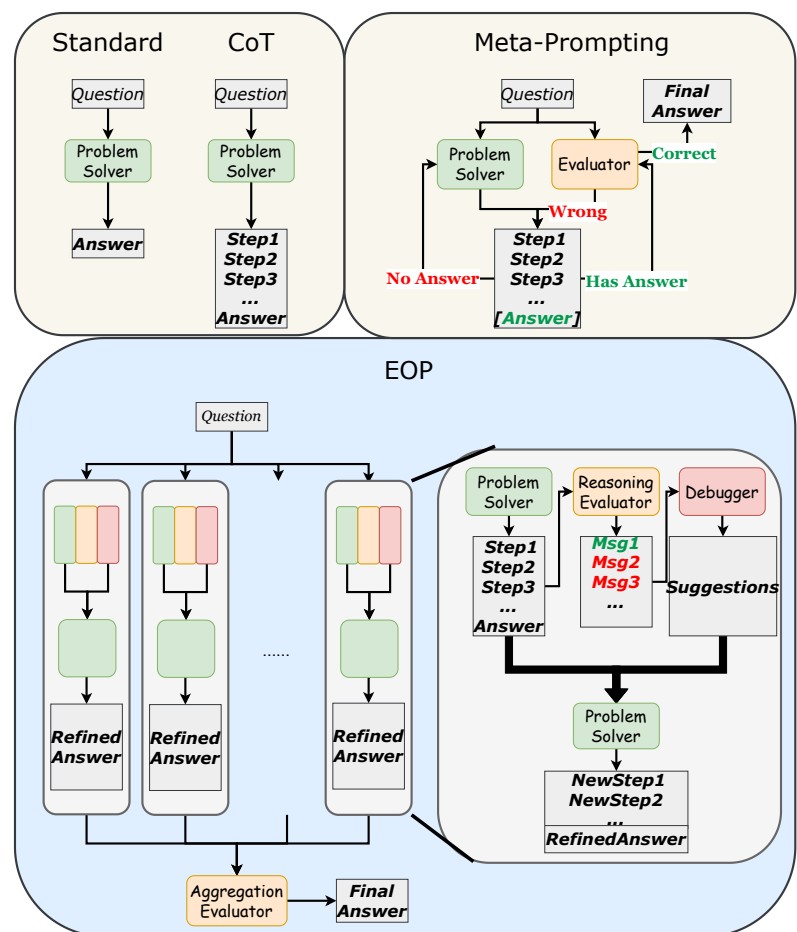

Figure 3: Overview of task-agnostic prompting methods. *Standard* refers to basic LLM-based problem-solving. *CoT* (Wei et al., 2022) prompts step-by-step thinking, while *Meta-prompting* (Suzgun & Kalai, 2024) refines answers using multiple rounds. Our *EOP* framework aggregates multiple trials with two evaluators: *Aggregation* and *Reasoning*.

**Task-agnostic Answer-refining:** Meta-prompting (Suzgun & Kalai, 2024) eliminates manual graph construction by allowing a meta-LLM to guide reasoning or call upon specialized expert LLMs. It also incorporates multi-agent debate (Du et al., 2023) and self-reflection (Shinn et al., 2023), making it state-of-the-art in many reasoning tasks.

**Solution Caching:** Buffer-of-Thoughts (Yang et al., 2024), on the other hand, aggressively caches solutions for reuse in similar tasks, but this method depends heavily on the quality of the cached solution and it introduces hard-coding, such as using the `chess` Python library for "Checkmate in One" problems.

## 3 METHODOLOGY

Figure 3 demonstrates an overview of EOP prompting pipeline. Our method introduces two key changes from other prompting methods based on GPT-4: the use of multiple trials (breadth-first) instead of multiple rounds (depth-first) and the incorporation of a reasoning evaluator. As described in Section 3.1, the problem solver powered by a small LLM generates multiple responses for each question, and an aggregation evaluator selects the best answer from these responses. This approach is crucial for improving overall accuracy. The second key change is the reasoning evaluator (Sec-

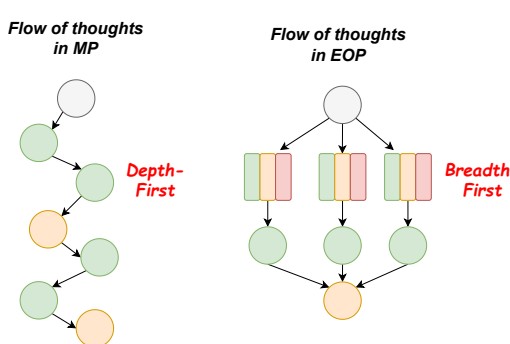

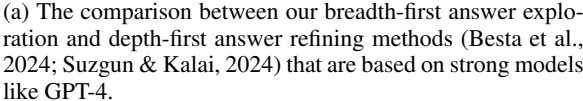

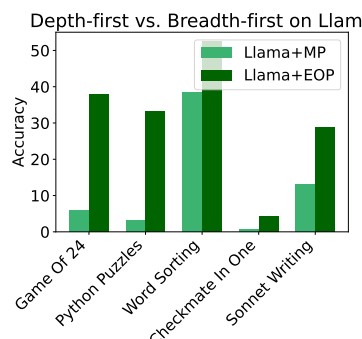

(a) The comparison between our breadth-first answer exploration and depth-first answer refining methods (Besta et al., 2024; Suzgun & Kalai, 2024) that are based on strong models like GPT-4.

(b) The comparisons of Llama3.1-8B (Meta, 2024) powered with meta-prompting (MP) (Suzgun & Kalai, 2024) and that powered by EOP. The benchmarks are described in Section 4

Figure 4: Comparisons between depth-first answer refining and breadth-first retrying.

tion 3.3), which evaluates the step-by-step reasoning process, rather than just the final answer. This evaluator processes the reasoning steps, and a debugger (Section 3.4) provides feedback to help refine the original answer. Although multiple iterations of refinement are possible, often one refinement pass suffices to improve the solution.

For tasks that can be converted into coding problems, such as Python tasks, both the reasoning and aggregation evaluators are Python interpreters, evaluating the reasoning process and final answer respectively (Section 3.2). Importantly, while the reasoning evaluator analyzes the full reasoning process, the aggregation evaluator only receives the final answer to avoid being misled by incorrect intermediate reasoning.

Since all tasks rely on proper evaluation, correct answer formatting is essential. To ensure that the LLM-generated responses are in the correct format, we utilize a hierarchical answer formulator (Section 3.5) to extract and structure the answers properly, preventing false rejections due to formatting issues.

### 3.1 MULTIPLE TRIALS

As shown in Figure 2, the accuracy of problem-solving continues to increase when conducting repeated trials, assuming we have an oracle evaluator. In contrast, methods such as ToT, GoT, and Meta-Prompting also use multiple rounds of reasoning. However, these approaches fail to significantly improve the performance of standard small LLMs, as demonstrated in Figure 4b. We suspect that this is because small LLMs are easily misled by incorrect answers during the refining process. This issue is discussed further in Section 3.2.

Given this, we opt to run small LLMs in multiple *independent rounds* using a *breadth-first* approach rather than adopting *depth-first* answer-refining methods, which are better suited for powerful models like GPT-4. The distinction between these approaches is illustrated in Figure 4a.

### 3.2 AGGREGATION EVALUATOR

As discussed in Section 1, small LLMs have the potential to produce correct answers across multiple trials. However, it is often challenging for these models to identify which of their generated answers is correct. For tasks that can be transformed into Python problems, a Python interpreter serves as a reliable oracle. In such cases, if one of the trial answers is correct, the interpreter will accurately validate the result.

For other tasks, the LLM must act as its own evaluator, responsible for assessing the correctness of its outputs. For example, in the Game of 24, evaluating whether an expression equals 24 is relatively

straightforward. However, small LLMs can still make evaluation errors, as shown in the following example:

```
Input:
2, 4, 4, 3

Output:
1. 2 + 4 = 6; 2. 6 * 4 = 24.
Each number of the input is used once.
From this analysis, (2 + 4) * 4 is considered an answer to
the question.

Evaluation:
The result is deemed correct, and each number is used once.
```

In this instance, the small LLM incorrectly evaluates the expression by assuming that each number is used only once, even though the number 3 is missing. Our tests showed that regardless of how the prompt is adjusted (e.g., through reasoning prompts (Qiao et al., 2022) or few-shot learning (Reynolds & McDonell, 2021)), there is always a risk that the LLM will misjudge such conditions. As a result, the evaluation accuracy is limited by the inherent weaknesses of small LLMs.

This limitation was also observed in more comprehensive experiments with the metric *evaluation accuracy* measured for the Game of 24 and Word Sorting as shown in Section 4.3.

### 3.3 REASONING EVALUATOR

#### 3.3.1 PYTHON PUZZLES

In contrast to previous work (Yao et al., 2024; Besta et al., 2024), our goal is to design a framework that is *task-agnostic*, eliminating the need for custom reasoning processes for each ad-hoc task. For instance, Meta-prompting (Suzgun & Kalai, 2024) introduces a pipeline with complex instructions, refining answers recursively and evaluating with a powerful model. However, applying such techniques to small LLMs often fails or even harms performance.

To create a generalizable reasoning process for small LLMs, we use Python Puzzles as a key example. Python problem-solving is widely applicable, and many other tasks, like Game of 24, can be transformed into Python problems. Instead of merely giving the LLM a wrong answer and expecting it to correct itself, EOP leverages the evaluator to provide more detailed feedback on why an answer is wrong, avoiding potential misleading information from the previous response. The *problem solver* first attempts to solve the task step by step, with the *reasoning evaluator* then assessing the correctness of each step in the reasoning process.

In Python Puzzles, maintaining syntactic correctness while solving a problem step-by-step can be difficult. To address this, we instruct the problem solver to insert `assert` statements at key points to verify partial results. If an assertion fails, the error, along with any other runtime errors, is reported to the *debugger* agent, which helps identify both logical and runtime issues.

For example, in Figure 5, we showcase a full reasoning flow from question to final answer. The task requires returning a list of even indices from a given list, with odd indices assigned a constant value. The assertions first verify the total length of the output list, then check whether the even-indexed elements match the input, and finally confirm that odd-indexed elements are equal to the constant value. The reasoning evaluator identifies an error in the second assertion, enabling downstream agents in the pipeline to refine the solution based on accurate feedback.

By adopting this approach, the LLM can pinpoint the specific part of the reasoning process responsible for the incorrect outcome, significantly reducing the risk of being misled by earlier wrong answers.

#### 3.3.2 GAME OF 24

Game of 24 is a task that can easily be converted into a Python program. In doing so, EOP is able to outperform all existing methods, including those that are *task-aware*, as demonstrated in

### Question Initializer ❶

Given a Python function "sat", the goal is to find an input or a set of inputs that makes "sat" return "True" and then include your input inside a function called "sol()".

**Insert assertion into the code to help verify the intermediate results**

```python
def sat(li: List[int], nums=[39, 94, 99, 46, 93], sep=25):
    return li[::2] == nums and li[1::2] == [sep] * (len(nums) - 1)
```

### Problem Solver ❷

```python
def sol():
    # Determine the correct input for the "sat" function
    nums = [39, 94, 99, 46, 93]
    sep = 25
    li = nums + [sep] * (len(nums) - 1)
    assert len(li) == len(nums) + (len(nums) - 1), "The length of li should be equal to the length
of nums plus (len(nums) - 1)"
    assert li[::2] == nums, "Even-indexed elements of li should be equal to nums"
    assert li[1::2] == [sep] * (len(nums) - 1), "Odd-indexed elements of li should be equal to sep
repeated (len(nums) - 1) times"
    return li
```

### Reasoning Evaluator ❸

An error occurred at line 11 of code
**Even-indexed elements of li should be equal to nums**

[39, 94, 99, 46, 93, 25, 25, 25, 25]

### Debugger ❹

### Possible Reasons for the Error
* The error occurs because the even-indexed elements of "li" are not equal to "nums".
* **This could be due to the fact that the list "li" is created by concatenating "nums" with "sep" repeated (len(nums) - 1) times, but the even-indexed elements of the resulting list are not equal to "nums".**
* To fix this error, the programmer needs to ensure that the even-indexed elements of "li" are equal to "nums" by modifying the creation of the list "li".

### Code Review
* The code is well-structured and uses assertions to handle edge cases.
* However, the assertion at line 11 fails, indicating that the even-indexed elements of "li" are not equal to "nums".
* To fix this error, the programmer needs to modify the creation of the list "li" to ensure that the even-indexed elements are equal to "nums".

### Problem Solver ❺

```python
def sol():
    # The even-indexed elements of the list should be equal to the elements of the list "nums"
    # The odd-indexed elements of the list should be equal to the element "sep" repeated (len(nums)
- 1) times
    # So, we create a list that meets these conditions
    li = [39, 25, 94, 25, 99, 25, 46, 25, 93]

    # Check if the length of "li" is equal to the length of "nums" plus (len(nums) - 1)
    assert len(li) == len([39, 94, 99, 46, 93]) + (len([39, 94, 99, 46, 93]) - 1), "Length of li is
incorrect"

    # Check if the even-indexed elements of "li" are equal to "nums"
    assert li[::2] == [39, 94, 99, 46, 93], "Even-indexed elements of li are not equal to nums"

    # Check if the odd-indexed elements of "li" are equal to "sep" repeated (len(nums) - 1) times
    assert li[1::2] == [25] * (len([39, 94, 99, 46, 93]) - 1), "Odd-indexed elements of li are not
equal to sep repeated (len(nums) - 1) times"

    # Return the input that makes "sat" return "True"
    return li
```

### Final Evaluator ❻

[39, 25, 94, 25, 99, 25, 46, 25, 93]
True

Figure 5: A demo of a successfully corrected Python Puzzle. The *Reasoning Evaluator* detects an assertion error and reports it to the *Debugger*, which identifies the error source and provides feedback to the *Problem Solver*.

Table 1. However, this approach does not evaluate the arithmetic reasoning capability within our EOP framework, as it predominantly tests Python coding skills, which overlap with tasks like Python

Table 1: Comparison of methods for solving Game of 24. Llama is short for Llama3.1-8B (Meta, 2024). ToT (Yao et al., 2024) and GoT (Besta et al., 2024) require the user to build a specific tree/graph structure for the thought. '-' represents *not supported*: Meta-prompting (MP) (Suzgun & Kalai, 2024) is task-agnostic, and it has both versions of using and not using a Python interpreter; While ToT and GoT are task-specific, and do not need the help of Python interpreter. EOP with Python is able to outperform all these methods.

| Frameworks | Task-specific | w/o Python | w/ Python |
|---|---|---|---|
| GPT-4 + ToT | 74.0% | - | - |
| GPT-4 + GoT | 73.2% | - | - |
| GPT-4 + MP | - | 11.0% | 67.0% |
| Llama + EOP | - | **37.8%** | **78.6%** |

Puzzles. To specifically assess arithmetic reasoning, we prohibit the use of the Python interpreter for the Game of 24 task. Instead, the *problem solver* generates a possible solution step by step, with a general *reasoning evaluator* verifying each step. Finally, the small LLM serves as an aggregation evaluator, as described in Section 3.2.

### 3.3.3 WORD SORTING

For the Word Sorting task, the evaluation can be quite challenging, as demonstrated in Section 4.3, where the accuracy of the aggregation evaluator hovers around 50%. Similar difficulties are also encountered with the reasoning evaluator. Although we could make the reasoning evaluator specific to the Word Sorting task, such as instructing it to compare two adjacent words at a time to simplify the evaluation, this approach introduces two issues: 1) it is not task-agnostic, and 2) even with this modification, the task remains beyond the capabilities of small LLMs, and no improvement in accuracy was observed using this method. Consequently, for Word Sorting and all tasks other than Python-related problems, we employ a general reasoning evaluator, as described in Section 3.3.2.

### 3.4 DEBUGGER

Reasoning evaluators assist in analyzing errors in previous answers. To streamline the incorporation of error analysis into the problem solver, we introduce a *debugger* that converts error messages into actionable suggestions. These suggestions, along with the original question and the prior answer, are then fed back to the problem solver. In this way, the connection between the errors and the solution refinement is smoother. We use the *Debugger* specifically for Python-related tasks, because Python interpreters only provide error messages without analysis. Debugger in other tasks simply passes the outputs of the reasoning evaluator.

### 3.5 HIERARCHICAL ANSWER FORMULATOR

Due to the reliance on the evaluation process, precise answer formulation is crucial, especially for Python tasks, as formatting errors can lead to compilation failures. To ensure correct answer formatting, we employ a hierarchical answer formulation unit: first, we attempt to extract the answer using predefined rules. If this fails, we utilize a small LLM to generate the answer in the specified format, followed by another attempt with the rule-based extractor. This process is repeated a limited number of times or until the correct format is obtained.

### 3.6 ADDRESSING THE KEY QUESTIONS

This research provides clear answers to the questions posed in Section 1 as follows:

- Applying answer-refining methods (Suzgun & Kalai, 2024) designed for models like GPT-4 does not yield significant improvements for small LLMs. These smaller models are more prone to being misled by incorrect answers during refinement (Section 3.1).
- While small LLMs can generate correct answers through multiple trials, the challenge lies in accurately selecting the correct answer from among the candidates.

Table 2: End-to-end experiments on different benchmarks, baselines, and small models. CoT (Wei et al., 2022) represents the SOTA of the chain of thoughts. MP is short for meta-prompting (Suzgun & Kalai, 2024). Llama represents Llama3.1-8B (Meta, 2024) and Qwen means Qwen2.5-7B models (Alibaba, 2024). Experiments are conducted on tasks: Game of 24 (Yao et al., 2024), Python Puzzles (Schuster et al., 2021), Word Sorting (Srivastava et al., 2023), Checkmate in One (Srivastava et al., 2023), and Sonnet Writing (Suzgun & Kalai, 2024). Green boxes represent best of all, and Blue boxes represents best of small LLMs.

| Task | GPT4 | GPT4 + CoT | GPT4 + MP | Llama | Qwen | Llama + EOP | Qwen + EOP |
|---|---|---|---|---|---|---|---|
| Game of 24 | 3.0% | 11.0% | 11.0% | 5.3% | 11.0% | 37.8% | 28.6% |
| Python Puzzles | 31.1% | 36.3% | 45.8% | 9.8% | 9.9% | 33.2% | 24.1% |
| Word Sorting | 80.4% | 83.6% | 84.0% | 39.5% | 16.6% | 52.4% | 22.8 % |
| Checkmate in One | 36.4% | 32.8% | 57.2% | 4.1% | 4.2% | 4.4% | 4.4% |
| Sonnet Writing | 62.0% | 71.2% | 77.6% | 24.0% | 29.1% | 28.8% | 32.8% |

- Our methodology proposes a comprehensive framework specifically aimed at enhancing the problem-solving abilities of small LLMs. Its effectiveness is demonstrated and validated through the experiments detailed in Section 4.

## 4 EXPERIMENTS

### 4.1 SETUP

To evaluate the performance of small LLMs, we conduct experiments on two sets of models: Llama3.1-8B (Meta, 2024) and Qwen2.5-7B (Alibaba, 2024). For coding tasks, in order to fully leverage the strengths of fine-tuned models, we use Qwen2.5-7B-Coder for the Qwen series experiments.

For the benchmark, in addition to *Game Of 24*, *Python Puzzles*, and *Word Sorting*, we also include *Checkmate in One* (Srivastava et al., 2023), which is a particularly challenging problem for small LLMs, demonstrating their limitations. Another benchmark, *Sonnet Writing* (Suzgun & Kalai, 2024), requires generating a sonnet with a restricted rhyme scheme.

For baselines, we primarily compare our results with meta-prompting and CoT, as other methods, such as multi-agent debate and self-reflection, are encapsulated within meta-prompting. Task-specific techniques like ToT and GoT are not considered task-agnostic and are thus excluded.

### 4.2 END-TO-END EXPERIMENT RESULTS

Table 2 presents an overall comparison of various baselines, models, and benchmarks. Notably, despite utilizing a significantly weaker model in EOP, we are still able to outperform GPT and even GPT+MP in several tasks. However, EOP also has its limitations. For the task of Checkmate in One, we are unable to further enhance the model's performance, as evaluating the results poses a significant challenge for smaller LLMs, resulting in an evaluation precision of only 4%.

### 4.3 ABLATION STUDIES ON EVALUATORS

As summarized in Table 3. We found that when the reasoning process is hidden from the aggregation evaluator, the accuracy for both tasks improved. By developing oracle evaluators tailored for each task, we observed that evaluation accuracy increased when the reasoning process was made transparent to the aggregation evaluator. This confirms our intuition that exposing the wrong reasoning process can mislead evaluators to some extent.

To assess the effectiveness of the *reasoning evaluator* introduced in Section 3.3, we conducted experiments to compare the accuracy across three problem-solving tasks, both with and without the use

Table 3: Comparison of two approaches for applying aggregation evaluators: reasoning-process aware vs. reasoning-process agnostic (Agn.). Accuracy improves for both tasks when the reasoning process is hidden from the aggregation evaluator. We also analyze evaluation accuracy using a manually designed oracle evaluator, tailored for a single task. The experiments utilize Llama3.1-8B.

| Task | Accuracy | Eval. Acc. |
|---|---|---|
| Game of 24 | 18.4% | 88.5% |
| Game of 24 Agn. | **33.7%** | **97.6%** |
| Word Sorting | 37.2% | 50.6% |
| Word Sorting Agn. | **42.0%** | **52.2%** |

Table 4: Comparison of problem-solving accuracy with and without the *reasoning evaluator*. The reasoning evaluator significantly enhances accuracy by providing detailed feedback on the correctness of each reasoning step. Experiments utilize Llama3.1-8B.

| Task | Acc. w/o Reasoning Eval. | Acc. w/ Reasoning Eval. |
|---|---|---|
| Game of 24 | 33.7% | **37.8%** |
| Python Puzzles | 27.2% | **33.2%** |
| Word Sorting | 40.4% | **52.4%** |
| Checkmate in One | 4.2% | **4.4%** |
| Sonnet | 24.8% | **28.8%** |

Table 5: Comparison of accuracy with and without the *Debugger* in Python Puzzle. The *Debugger* is essential, as the absence of it leads to reduced accuracy from the reasoning evaluator.

| Task | Acc. w/o Reasoning Eval. & Debugger | Acc. w/ Reasoning Eval. & w/o Debugger | Acc. w/ Debugger & Reasoning Eval. |
|---|---|---|---|
| Python Puzzles | 27.2% | 24.1% | **33.2%** |

of reasoning evaluators. The results, presented in Table 4, demonstrate a significant improvement in accuracy when the reasoning evaluators are employed, which evaluate the *problem solver*'s reasoning process step-by-step. This confirms our hypothesis that small LLMs benefit from more detailed feedback on their errors, enabling them to refine incorrect answers more effectively. The difference between *aggregation evaluator* and *reasoning evaluator* is one of our major contributions.

We further verify the effectiveness of employing a debugger to analyze error messages from the Python interpreter, which serves as our reasoning evaluator. Unlike an LLM-based evaluator, the Python interpreter does not inherently provide detailed analysis of reasoning errors. Therefore, a dedicated debugger is required to extract this information. The results presented in Table 5 underscore the significance of utilizing this debugger for improving reasoning accuracy. This experiment also indicates that using a reasoning evaluator combined with a debugger can further improve an oracle evaluator.

## 5 CONCLUSION

In this work, we explored the Evaluation-Oriented Problem-Solving (EOP) framework to enhance small LLMs' problem-solving abilities. Our experiments showed that EOP allows small LLMs to achieve competitive performance across tasks like Python coding, word sorting, and mathematical reasoning. The use of reasoning evaluators significantly improved accuracy by enabling error correction and output evaluation. However, challenges remain, particularly in complex tasks like Checkmate in One. Future work will focus on refining the reasoning evaluator and further improving small LLMs' robustness in challenging tasks.

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

## A   ABLATION STUDY OF ACCURACY VS. NUMBER OF MULTIPLE TRIALS

The number of trials performed by small LLMs significantly impacts the accuracy of their final output, as illustrated in Figure 2. After applying the full EOP framework, the accuracy improvement follows a similar trend, as shown in Figure 6. Conducting 10 trials yields substantial accuracy gains across the three tasks, while increasing the number of trials to 30 shows diminishing returns. Users should consider adopting a moderate number of trials to strike a balance between accuracy and computational cost.

## B   MEASURE OF THE EOP EFFECTIVENESS ON DIFFERENT TASKS

EOP has focused on quantifying its effectiveness quantitatively. At first, the *evaluation accuracy of the aggregation evaluator* is proposed. For example, tasks like Python Puzzles achieve an evaluation accuracy of 100% due to the availability of reliable evaluation methods (e.g., a Python interpreter). Similarly, Game of 24 achieves 97.6% evaluation accuracy as shown in Table 3. However, tasks like Word Sorting, with only 52.2% evaluation accuracy, show smaller improvements in problem-solving accuracy under EOP.

To address a potential limitation of this evaluation accuracy, where a large number of true negatives might inflate the metric when the model generates excessive incorrect answers, we propose using the F1 score instead. This provides a more balanced measure by considering both precision and recall. After adopting the F1 score, the results are shown in Table 6.

This relationship is intuitive: if the aggregation evaluator can reliably determine the correctness of answers (reflected by high F1 scores), it is more likely to select the correct answer from the pool of candidates. EOP is designed to exploit this property by enhancing the quality of candidates through breadth-first multiple trials, reasoning evaluators, debuggers, and answer formatters.

To address when EOP is most effective, we propose two key considerations:

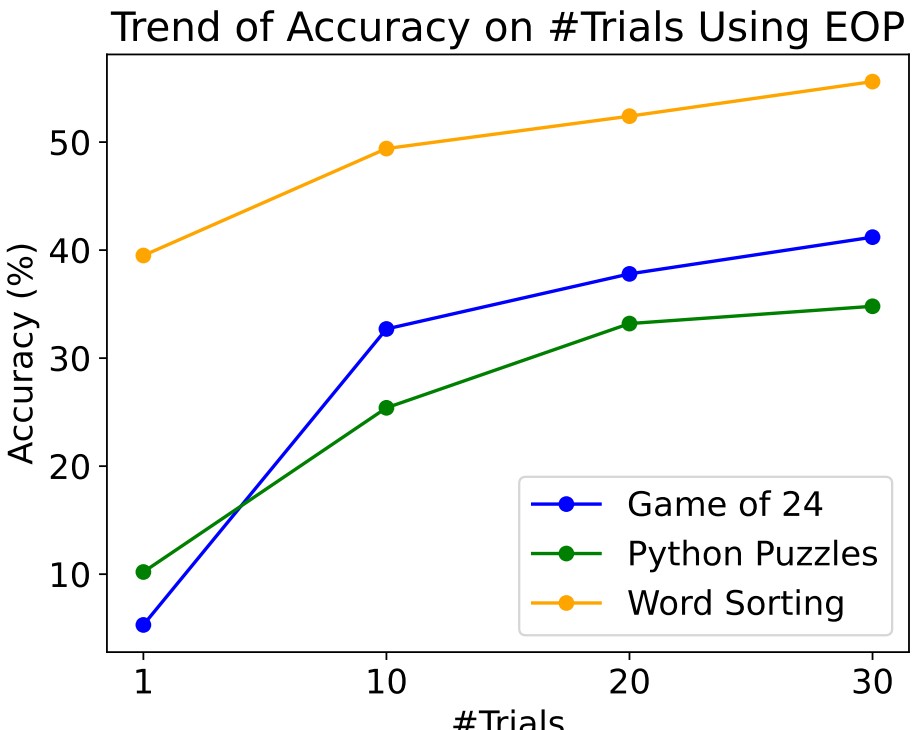

Figure 6: Ablation study of accuracy vs. number of trials. The accuracy improves marginally when the number of attempted trials increases.

| Task | Accuracy Improvement | Aggregation Evaluator F1 |
|---|---|---|
| Python Puzzles | +23.4% | - |
| Game of 24 | +32.5% | 0.83 |
| Word Sorting | +12.9% | 0.52 |
| Checkmate-in-One | +0.3% | 0.05 |

Table 6: EOP accuracy improvement and F1 scores of the aggregation evaluator for different tasks.

- **When evaluating a problem is easier than solving it:** In such cases, evaluation-based methods act like SAT-solvers for NP-hard problems, leveraging simpler evaluations to tackle complex problems.

- **Using F1 score of the aggregation evaluator as a predictor:** Users can assess the F1 score of the aggregation evaluator using sample questions from the target task. While there isn't a universal threshold (different models have different capabilities on different tasks), this measure provides a clearer indication of whether EOP can achieve significant accuracy improvements. For instance, tasks with low F1 scores like Checkmate-in-One (0.05) yield marginal improvements, while tasks with higher F1 scores like Game of 24 (0.83) benefit significantly.

