# OpenReview forum: "EOP: Unlocking Superior Problem Solving in Small LLMs"
_ICLR.cc/2025/Conference — Submitted to ICLR 2025_

### Official Review · Reviewer_S7f6 · 2024-10-25

**Soundness:** 2
**Presentation:** 3
**Contribution:** 2
**Rating:** 5
**Confidence:** 3

**Summary:**

This paper focuses on the application of Large Language Models (LLMs) with fewer than 10 billion parameters to problem-solving tasks. The existing small LLMs are usually unable to achieve high accuracy when dealing with problem-solving tasks, while lots of the existing researches usually focus on how to enhance the performance of large LLMs, ignoring the need for improving the accuracy of small LLMs. In this paper, the authors propose a novel framework, Evaluation-Oriented Problem-Solving (EOP), to enhance the ability of small LLMs to handle problem-solving tasks. With the help of the EOP framework, the performance of the existing small LLMs on the Python Puzzles and Game of 24 problems is further improved and is better than the standard GPT-4, indicating a new direction beyond the approach of increasing the capability of the LLMs by increasing the size of the model.

**Strengths:**

1. Instead of increasing the model size to improve the performance of LLMs, this paper designed an evaluation-based approach and implemented the Evaluation-Oriented Problem-Solving (EOP) framework for small LLMs. The research in this paper provides new guidance for improving the performance of LLMs.

2. The EOP framework designed by the author successfully improves the performance of small LLMs in problem-solving tasks such as Game of 24, Python Puzzles and Word Sorting. More encouragingly, it successfully outperforms the standard GPT-4 on Python Puzzles and Game of 24 tasks. The experimental results show that the reasoning evaluator plays an important role in improving the performance of small LLMs.

**Weaknesses:**

1. The motivation of this paper is a bit unclear. In practice, large language models ahcieve much better performance than small language models. Thus, what is the motivation for enhancing small language models? Why don't the authors further improve the performance of large language models?

2. The interpretation of the experimental results is too brief, so it is suggested that the author conduct a more detailed analysis of the experimental results.

3. The author mentioned in the paper that EOP mainly introduced two key changes -- the use of multiple trials (breadth-first) and the integration of a reasoning evaluator. In the ablation experiment part of Section 4.3, it seems that the ablation analysis of the improvement using multiple trials is not shown, and it is suggested to supplement the relevant part to make the experiment more complete.

4. There is a problem with the format of line 279. Appropriate adjustments are recommended.

**Questions:**

Please refer to all the comments listed in "Weaknesses".

---

> ### Author Response · Authors · 2024-11-21
>
> **[Q1]**: what is the motivation for enhancing small language models? Why don't the authors further improve the performance of large language models?
>
> **[A1]**: For the motivation of small LLMs, please refer to the answer for Reviewer 5J8o.
>
> ---
>
> **[Q2]**: The author mentioned in the paper that EOP mainly introduced two key changes -- the use of multiple trials (breadth-first) and the integration of a reasoning evaluator. In the ablation experiment part of Section 4.3, it seems that the ablation analysis of the improvement using multiple trials is not shown, and it is suggested to supplement the relevant part to make the experiment more complete.
>
> **[A2]**:
>
> We would like to clarify that EOP introduces three major components: *Multiple Trials*, *Aggregation Evaluator*, and *Reasoning Evaluator*, not just two as mentioned by the reviewer. The ablation studies for these components are included in the paper:
>
> 1. **Aggregation Evaluator**: Ablation results are shown in Table 3, which also examines whether the evaluator is aware of the reasoning process.
> 2. **Reasoning Evaluator**: Table 4 presents the ablation analysis for the Reasoning Evaluator, demonstrating its contribution to performance improvements.
> 3. **Multiple Trials**: Figure 2 provides ablation analysis for Multiple Trials in isolation (without involving evaluators) using "ideal accuracy" as the metric.
>
> If the reviewer is referring to ablation studies involving Multiple Trials alongside evaluators, we acknowledge that such an analysis is not explicitly provided. However, trends similar to those in Figure 2 can be expected. For instance, with different numbers of total trials, the accuracy for tasks like Python Puzzles and Word Sorting exhibits the following pattern:
>
> | Tasks           | 1    | 10   | 20   | 30   |
> |------------------|------|------|------|------|
> | Python Puzzles  | 10.2 | 25.4 | 33.2 | 34.8 |
> | Word Sorting    | 23.2 | 49.2 | 52.4 | 54.3 |
>
> To make the experimental results more complete, we have added more comprehensive analyses, including the interplay between Multiple Trials and evaluators, in the Appendix.
>
> ---
>
> **[Q3]**: The interpretation of the experimental results is too brief, so it is suggested that the author conduct a more detailed analysis of the experimental results.
>
> **[A3]**: Thank you for the suggestion. We acknowledge that the interpretation of the experimental results could benefit from more detailed analysis. To address this, we added more analysis and experiments in the Appendix section of the paper
>
> ---
>
> **[Q4]**: There is a problem with the format of line 279. Appropriate adjustments are recommended.
>
> **[A4]**: Thanks for the suggestion, we went through the paper and fixed the formatting issues and page boundaries.

---

### Official Review · Reviewer_5J8o · 2024-10-25

**Soundness:** 3
**Presentation:** 3
**Contribution:** 3
**Rating:** 6
**Confidence:** 3

**Summary:**

The paper studies challenges faced by small LLMs (fewer than 10 billion parameters) in problem-solving tasks, where they often achieve less than 10% accuracy. The authors investigated whether advanced prompting techniques designed for larger models, such as meta-prompting, can enhance the performance of smaller models, but find limited success. Then they propose a new framework called Evaluation-Oriented Problem-Solving (EOP), which focuses on tailored evaluation techniques and multiple trials to improve accuracy. The study demonstrates that small LLMs can perform better when using an oracle evaluator to select correct answers from multiple attempts. The paper presents experimental results showing that EOP can significantly enhance the problem-solving capabilities of small LLMs across various tasks, outperforming existing methods and even some larger models in specific scenarios.

**Strengths:**

1. The scientific flow of this paper is great. I like the way the authors first identified the issue with experiments and then proposed their method to address it. The paper is also well written what methodology, and results clearly stated. The figures also help readers understand the idea a lot.

2. The experiment results are encouraging. EOP significantly improved the performance of small LLMs, achieving a 2% higher accuracy on Python Puzzles compared to GPT-4 and a 27% improvement over state-of-the-art methods in the Game of 24.

**Weaknesses:**

1. The problem studied in the paper is important in that the size of LLMs can be important in specific scenarios. However, the motivation is not well established in the paper. It would be nice to have a paragraph to educate the readers on why small LLMs are preferred over large LLMs and why the challenging faced by small LLMs are critical.

**Questions:**

Why is using small LLMs critical?

---

> ### Author Response · Authors · 2024-11-21
>
> **[Q1]**: Why is using small LLMs critical?
>
> **[A1]**:
>
> Thanks for the valuable suggestion, we have added more justifications for the importance of optimizing small LLMs.
>
> ---
>
> **The Power of Small LLMs**
>
> Small LLMs are gaining traction due to their advantages in cost and energy efficiency, lower hardware requirements for self-deployment, and reduced domain adaptation costs, all while maintaining relatively high performance on various benchmarks such as reading comprehension and common-sense understanding [1, 2, 3]. Recent studies have shown that small LLMs can achieve performance comparable to larger models on many standard tasks, making them an attractive option for scenarios where resource constraints are critical.
>
> **Challenges in Reasoning Tasks**
>
> Despite their strengths, small LLMs struggle significantly in tasks requiring reasoning, such as math, logical problem-solving, and coding [3]. While large LLMs perform well in these areas, their high computational cost and energy requirements make them less feasible for many applications. This creates a performance gap in leveraging small LLMs for challenging problem-solving tasks.
>
> To address this gap, we introduce EOP, a framework specifically designed to enhance small LLMs' performance on reasoning-intensive tasks. This research represents an important and underexplored area, offering a pathway to balance cost efficiency and performance. By enabling small LLMs to tackle reasoning tasks effectively, EOP contributes to the broader goal of optimizing LLM applications for diverse use cases where both efficiency and effectiveness are crucial.
>
> ---
>
> [1] Bansal H, Hosseini A, Agarwal R, et al. Smaller, weaker, yet better: Training llm reasoners via compute-optimal sampling[J]. arXiv preprint arXiv:2408.16737, 2024.
>
> [2] Javaheripi M, Bubeck S, Abdin M, et al. Phi-2: The surprising power of small language models[J]. Microsoft Research Blog, 2023, 1: 3.
>
> [3] Dubey A, Jauhri A, Pandey A, et al. The llama 3 herd of models[J]. arXiv preprint arXiv:2407.21783, 2024.

---

### Official Review · Reviewer_bUFf · 2024-11-01

**Soundness:** 3
**Presentation:** 3
**Contribution:** 2
**Rating:** 5
**Confidence:** 4

**Summary:**

The paper proposes a method to improve problem solving for small LLMs,
leveraging repeated prompting. The authors describe their method and evaluate it
empirically, comparing to large commercial LLMs.

**Strengths:**

The paper tackles an interesting and timely problem. LLMs are very powerful, but
have prohibitive resource requirements. This restricts who can benefit from
them, and research to make them more accessible is very important.

**Weaknesses:**

The proposed method seems to work well in practice, but it is unclear how
general it is. The authors state that "with a reliable evaluator, small models
can achieve superior accuracy" but also that "the evaluation accuracy is limited
by the inherent weaknesses of small LLMs". The empirical results are also
somewhat mixed; in some cases the proposed methodology results in only small
changes. It is unclear why this happens and under what circumstances the
proposed methodology is likely to produce substantially better results. This is
an important question that the paper does not consider.

Further, the problems considered in the paper are relatively easy -- how does
the proposed approach fare on difficult tasks, in particular ones that users are
likely interested in? In particular, are small LLMs capable of evaluating the
solutions for more difficult tasks with reasonable accuracy? If they are not, it
seems that the proposed approach would not work at all anymore, limiting its
applicability.

**Questions:**

How would the approach fare with more difficult problems?

Update after responses: Thank you for your responses.

---

> ### Author Response · Authors · 2024-11-21
>
> **[Q1]**: It is unclear why this happens and under what circumstances the proposed methodology is likely to produce substantially better results. This is an important question that the paper does not consider.
>
> **[A1]**: The conditions under which the proposed methodology produces substantially better results are indeed a key focus of our analysis, and we address this in our fourth contribution (page 3). Specifically, we analyze how the evaluation difficulty of different reasoning tasks influences the effectiveness of the EOP framework.
>
> As discussed in the last paragraph of page 2 and illustrated in Figure 1, tasks such as Python Puzzles and Game of 24 are inherently easier to evaluate. Python Puzzles benefit from the availability of a Python interpreter, and evaluating whether an expression equals 24 in Game of 24 is straightforward for LLMs. This alignment between task characteristics and EOP's evaluator-centric design leads to significant performance improvements in these tasks.
>
> In contrast, tasks like Word Sorting present greater evaluation complexity, as determining whether a list of words is correctly sorted needs multiple steps of thought, making it less intuitive for small LLMs. This mismatch results in smaller accuracy improvements, highlighting the impact of evaluation difficulty on EOP's effectiveness.
>
> This evaluation difficulty vs. accuracy improvement relationship is further elaborated in Section 4.2. By quantitatively addressing this relationship, we demonstrate that EOP is not just a general-purpose method but one that is specifically optimized to tackle challenges in problem-solving tasks with varying evaluation complexities.
>
> ---
>
> **[Q2]**: In particular, are small LLMs capable of evaluating the solutions for more difficult tasks with reasonable accuracy?
>
> **[A2]**:
> We acknowledge that small LLMs have inherent limitations, and even with EOP, they may struggle to perform well on particularly difficult tasks. However, this does not diminish the value of researching methods to improve their accuracy, especially on the tasks of math, reasoning, and coding, which are already well-known “difficult” tasks for small LLMs [1].
>
> To clarify, the inapplicability of small LLMs to certain challenging tasks does not imply that EOP is ineffective. For example, small LLMs struggle with Checkmate-in-one tasks, even when enhanced by EOP. However, EOP demonstrates significant success in coding problems (shown in Table 4), enabling small LLMs to achieve superior performance, even outperforming GPT-4 in some cases (Figure 1). Importantly, coding problems are highly practical and commonly encountered, making EOP's impact in this domain particularly valuable.
>
> Moreover, tasks like Checkmate-in-one can be reframed to align better with the strengths of small LLMs. By converting the problem into a coding task using tools like the Python chess library, EOP enables small LLMs to leverage their coding capabilities, resulting in reasonable accuracy improvements.
>
> ---
>
> [1] Dubey A, Jauhri A, Pandey A, et al. The llama 3 herd of models[J]. arXiv preprint arXiv:2407.21783, 2024.

---

> > ### Comment · Reviewer_bUFf · 2024-11-21
> >
> > Thank you for your reply. The answers to when your methodology will produce better results then seem to hinge on what evaluation method is available, in particular something reliable and easy to run like a Python interpreter will lead to better results. But what when this is not the case (which I believe would be most practical applications)? What problem characteristics (rather than available evaluation methodologies) make a problem amenable to the methodology you propose?

---

> ### Author Response · Authors · 2024-11-25
>
> Thank you for your insightful question. It indeed highlights a critical aspect: understanding the conditions under which evaluation-based methodologies like EOP work effectively.
>
> Our research has focused on quantifying this "effectiveness" through experiments, particularly in Table 3, where we evaluate the "evaluation accuracy" of the aggregation evaluator for different tasks. For example, tasks like Python Puzzles achieve an evaluation accuracy of 100% due to the availability of reliable evaluation methods (e.g., a Python interpreter). Similarly, Game of 24 achieves 97.6% evaluation accuracy. However, tasks like Word Sorting, with only 52.2% evaluation accuracy, shows smaller improvements in problem-solving accuracy under EOP.
>
> To address a potential limitation of "evaluation accuracy," where a large number of true negatives might inflate the metric when the model generates many incorrect answers, we propose using the F1 score instead. This provides a more balanced measure by considering both precision and recall. After adopting the F1 score, the results are as follows:
>
> | **Tasks**            | **Accuracy Improvement** | **Aggregation Evaluator F1** |
> |-----------------------|--------------------------|-------------------------------|
> | Python Puzzles        | +23.4%                  | -                             |
> | Game of 24            | +32.5%                  | 0.83                          |
> | Word Sorting          | +12.9%                  | 0.52                          |
> | Checkmate-in-One      | +0.3%                   | 0.05                          |
>
> This relationship is intuitive: if the aggregation evaluator can reliably determine the correctness of answers (reflected by high F1 scores), it is more likely to select the correct answer from the pool of candidates. EOP is designed to exploit this property by enhancing the quality of candidates through breadth-first multiple trials, reasoning evaluators, debuggers, and answer formatters.
>
> To address when EOP is most effective, we propose two key considerations:
>
> 1. **When evaluating a problem is easier than solving it:**
>    In such cases, evaluation-based methods act like SAT-solvers for NP-hard problems, leveraging simpler evaluations to tackle complex problems.
>
> 2. **Using F1 score of the aggregation evaluator as a predictor:**
>    Users can assess the F1 score of the aggregation evaluator using sample questions from the target task. While there isn’t a universal threshold (different models have different capabilities on different tasks), this measure provides a clearer indication of whether EOP can achieve significant accuracy improvements. For instance, tasks with low F1 scores like Checkmate-in-One (0.05) yield marginal improvements, while tasks with higher F1 scores like Game of 24 (0.83) benefit significantly.
>
> Sometimes, this "evaluation difficulty" can be quite obvious: Checkmate in One requires checking step-by-step a long sequence of SAN (Standard Algebraic Notation), e.g. **1. d4 d5 2. Nf3 Nf6 3. e3 a6 4. Nc3 e6 5. Bd3 h6 6. e4 dxe4 7. Bxe4 Nxe4 8. Nxe4 Bb4+ 9. c3 Ba5 10. Qa4+ Nc6 11. Ne5 Qd5 12. f3 O-O 13. Nxc6 bxc6 14. Bf4 Ra7 15. Qb3 Qb5 16. Qxb5 cxb5 17. a4 bxa4 18. Rxa4 Bb6 19. Kf2 Bd7 20. Ke3 Bxa4 21. Ra1 Bc2 22. c4 Bxe4 23. fxe4 c5 24. d5 exd5 25. exd5 Re8+ 26. Kf3 Rae7 27. Rxa6 Bc7 28. Bd2 Re2 29. Bc3 R8e3+ 30. Kg4 Rxg2+ 31. Kf5** and keep the chess state correct hiddenly for every step, which is extremely difficult, especially for small LLMs.
>
> To further enhance the paper, we have extended the evaluation accuracy and F1 score analysis and included these results, along with a detailed analysis, in the Appendix.

---

### Official Review · Reviewer_AK4r · 2024-11-04

**Soundness:** 2
**Presentation:** 4
**Contribution:** 1
**Rating:** 5
**Confidence:** 2

**Summary:**

This paper investigates whether Evaluation-Oriented Problem-Solving can enhance the problem-solving capabilities of small LLMs. By comparing the performance of large and small LLMs using CoT, GoT and MP methods, the study concludes that small LLMs perform worse than large models in several types of problem-solving tasks. Therefore, innovative methods are needed to address this performance gap in small models. The paper demonstrates through experiments that the EOP approach improves the problem-solving performance of small LLMs.

**Strengths:**

1. The writing is clear and easy to understand.
2. In the proposed framework, the paper uses multiple trials (breadth-first) instead of multiple rounds (depth-first) and incorporates a reasoning evaluator, effectively enhancing the reasoning and synthesis capabilities of small models.

**Weaknesses:**

1. Evaluation-oriented approaches in LLMs are more and more studied in many learning tasks. This paper seems the first one that studies evaluation-oriented approach for problem solving. They aim to propose a small LLMs specific approach. How does EOP perform for large models? The paper does not conduct experiments for it.
2. The authors point out that the paper includes four main contributions: understanding failure modes, emphasizing the role of evaluators, introducing a novel evaluation-based prompting method, and addressing varying levels of evaluator expertise. It seems that the first three contributions are consensus in today’s study. The novelty of the contributions is limited.

**Questions:**

1. The evaluation-oriented approach is anticipated to positively influence reasoning tasks for large models, as supported by empirical evidence from similar studies. The paper validates the effectiveness of the EOP approach for small LLMs, which aligns with intuitive expectations. The novelty of this work is unclear.
2. If the paper intends to focus on an evaluation-oriented approach specifically for problem-solving, I expect an analysis of the characteristics of problem-solving tasks and how these can be addressed within the EOP framework. This would help clarify the novelty and contributions of the paper.

---

> ### Author Response · Authors · 2024-11-21
>
> **[W1]**: How does EOP perform for large models? The paper does not conduct experiments for it.
>
> **[A1]**: This work is specifically motivated by the observation that existing reasoning prompting methods designed for larger LLMs do not translate effectively to smaller LLMs. Our primary focus is to understand and address this gap by identifying the unique challenges faced by smaller models and developing the EOP method to significantly improve their problem-solving capabilities (3 questions in page 2). The experimental scope and contributions of this paper are intentionally limited to small LLMs, as larger models have already been extensively studied and optimized in prior work. Exploring EOP’s applicability to larger models could be an interesting direction for future research, but it falls outside the scope of this study.
>
> ---
>
> **[W2]**: The authors point out that the paper includes four main contributions: understanding failure modes, emphasizing the role of evaluators, introducing a novel evaluation-based prompting method, and addressing varying levels of evaluator expertise. It seems that the first three contributions are consensus in today’s study. The novelty of the contributions is limited.
>
> **[A2]**:
> While the concepts of failure modes, the role of evaluators, and evaluation-based prompting have been explored to some extent in prior work, this paper makes novel contributions by addressing these aspects specifically in the context of small LLMs, which have unique challenges not observed in larger models.
>
> - **Failure Modes**: The failure modes we identified are specific to small LLMs and have not been discussed in prior work. These modes arise because methods successful in larger LLMs fail when applied to smaller models. We present a detailed analysis of why these methods don’t work in this setting, providing new insights into small LLM limitations in Section 3.1.
> - **Role of Evaluators**: Our contribution goes beyond simply emphasizing that "evaluators are important." We propose a breadth-first organization of evaluators for small LLMs, contrasting with the depth-first structures typically used in larger models. Additionally, we introduce finer-grained evaluator distinctions: (1) reasoning evaluators that assess intermediate reasoning steps and (2) aggregation evaluators that judge the correctness of the final output. This dual-layered approach addresses the vulnerability of small LLMs to reasoning errors that propagate to incorrect final answers. (Section 3.2 and Section 3.3)
> - **Evaluation-Based Prompting**: We provide a concrete and novel implementation of evaluation-based prompting tailored to small LLMs. For instance, in coding tasks, we introduce assert-based reasoning evaluators that leverage Python interpreters to conduct fine-grained evaluations of intermediate reasoning steps. This method results in significant improvements in accuracy, as demonstrated in Section 3.3.1. To the best of our knowledge, this is the first approach concerning how to fully utilize the Python interpreter for reasoning process evaluation, making it a unique contribution to the field.
> Our work introduces novel insights and practical solutions specifically for small LLMs, bridging a critical gap in the current understanding of these models and their capabilities.

---

> ### Author Response · Authors · 2024-11-21
>
> **[Q1]**: The evaluation-oriented approach is anticipated to positively influence reasoning tasks for large models, as supported by empirical evidence from similar studies. The paper validates the effectiveness of the EOP approach for small LLMs, which aligns with intuitive expectations. The novelty of this work is unclear.
>
> **[A1]**: While it may seem intuitive that evaluators can improve reasoning performance, this oversimplifies our contributions. As demonstrated in Figure 1, simply applying evaluator-based methods as done in prior studies on large LLMs fails to significantly improve accuracy for small LLMs. This raises two critical questions that our paper addresses:
>
> - **Why do current evaluator-based methods fail for small LLMs?**
> - **How can evaluators be adapted to work effectively for small LLMs?**
>
> These are the core contributions of this work. We provide an in-depth analysis of the failure modes unique to small LLMs and propose specific adaptations, such as breadth-first evaluator architectures (Figure 4 (a) and (b)) and finer-grained evaluator distinctions (reasoning evaluators (Section 3.3) vs. aggregation evaluators (Section 3.2)), that overcome these limitations.
>
> Furthermore, as shown in Table 2, our EOP approach significantly outperforms established methods like Chain of Thought (CoT) and Meta-Prompting (MP) on small LLMs. This demonstrates that the success of our method is not a straightforward extension of existing techniques but instead stems from addressing non-intuitive challenges specific to small LLMs.
>
> ---
>
> **[Q2]**: If the paper intends to focus on an evaluation-oriented approach specifically for problem-solving, I expect an analysis of the characteristics of problem-solving tasks and how these can be addressed within the EOP framework. This would help clarify the novelty and contributions of the paper.
>
> **[A2]**: The relationship between task characteristics and their compatibility with the EOP framework is indeed central to our study and is explicitly addressed as our fourth contribution. Specifically, we analyze how the varying levels of evaluation difficulty inherent to different tasks influence the effectiveness of EOP.
>
> As discussed in the last paragraph of page 2 and illustrated in Figure 1, tasks such as Python Puzzles and Game of 24 are inherently easier to evaluate. Python Puzzles benefit from the availability of a Python interpreter, and evaluating whether an expression equals 24 in Game of 24 is straightforward for LLMs. This alignment between task characteristics and EOP's evaluator-centric design leads to significant performance improvements in these tasks.
>
> In contrast, tasks like Word Sorting present greater evaluation complexity, as determining whether a list of words is correctly sorted needs multiple steps of thought, making it less intuitive for small LLMs. This mismatch results in smaller accuracy improvements, highlighting the impact of evaluation difficulty on EOP's effectiveness.
>
> This evaluation difficulty vs. accuracy improvement relationship is further elaborated in Section 4.2. By quantitatively addressing this relationship, we demonstrate that EOP is not just a general-purpose method but one that is specifically optimized to tackle challenges in problem-solving tasks with varying evaluation complexities.

---

### Meta-Review · Area_Chair_sXfJ · 2024-12-20

**Metareview:**

The paper focuses on small LLMs (SML),  motivated by the observation that existing reasoning prompting methods designed for larger LLMs do not translate effectively to smaller LLMs.   An Evaluation-Oriented Problem-Solving (EOP)  module is proposed, dedicated to SML, involving  reasoning evaluators that assess intermediate reasoning steps and  aggregation evaluators that judge the correctness of the final output.

The results show significant improvements on some taskscompared to large language models.

**Additional Comments On Reviewer Discussion:**

The rebuttal did an excellent job in explaining why small LLMs matter. The approach, notably based on the use of evaluators such as  a Python interpreter, focuses on assessing both the chain of steps and the final results. The experiments show improvement on CoT and MP.

Overall, the approach seems very promising.The reviewers' concern are that the level of difficulty of the considered problems might be insufficient to establish its merits;  the results are not yet overwhelmingly convincing and their discussion needs be strengthened.

The area chair warmly encourages the authors to continue  this line of research.

---

### Decision · Program_Chairs · 2025-01-22

Reject